# Association between Sugar Intake and Intima Media Thickness as a Marker for Atherosclerosis: A Cross-Sectional Study in the Malmö Diet and Cancer Study (Sweden)

**DOI:** 10.3390/nu13051555

**Published:** 2021-05-05

**Authors:** Esther González-Padilla, Suzanne Janzi, Stina Ramne, Camilla Thuneland, Yan Borné, Emily Sonestedt

**Affiliations:** Nutritional Epidemiology, Department of Clinical Sciences Malmö, Lund University, SE-214 28 Malmö, Sweden; suzanne.janzi@med.lu.se (S.J.); stina.ramne@med.lu.se (S.R.); camilla.thuneland@hotmail.com (C.T.); yan.borne@med.lu.se (Y.B.); emily.sonestedt@med.lu.se (E.S.)

**Keywords:** sugar intake, added sugar, free sugar, total sugar, sugar-rich foods, sugar-sweetened beverages, SSB, intima media thickness, IMT, atherosclerosis

## Abstract

It has been suggested that sugar intake may play a role in the development of atherosclerosis. However, studies on this matter are lacking. Intima media thickness (IMT) is a well-established measurement of subclinical atherosclerosis. This study aimed to investigate the cross-sectional association between sugar intake (i.e., added, free and total sugar and sugar-rich foods and beverages) and IMT. Our study comprised 5269 individuals (45–73 years, 40% men) of the Malmö Diet and Cancer Study, a population-based cohort conducted in Sweden with data collected from 1991 to 1994. Measurements of IMT were performed with B-mode ultrasound at the right common carotid artery (IMTcca) and the bifurcation of the carotids (IMTbif). Dietary intake was estimated using a combination of a 7-day food record, diet questionnaire and interview. After adjusting for methodological, lifestyle and dietary confounders, no statistically significant associations were observed for any of the sugar intake variables and IMT. For example, added sugar intake presented no significant linear association with IMTcca or IMTbif (P_trends_: IMTcca 0.81 for men and 0.98 for women and IMTbif 0.20 for men and 0.40 for women). In conclusion, we found no clear association between sugar intake and IMT measurements in this study.

## 1. Introduction

Atherosclerosis is a complex progressive condition that starts early on in life due to the accumulation of cholesterol and other fatty materials on the walls of arteries forming plaques that narrow their diameter. Atherosclerosis is considered the main underlying process behind many cardiovascular diseases (CVDs) including ischaemic heart disease, myocardial infarction and cerebrovascular stroke [1]. CVD remains at the forefront of the list of major causes of morbidity and mortality worldwide. Yet, many known risk factors for CVD and atherosclerosis are modifiable, including tobacco use, alcohol abuse, physical inactivity and an unhealthy diet [1]. A healthier diet could have beneficial effects by affecting the atherosclerotic progression [2]. The measurement of the artery wall or intima media thickness (IMT) of the carotids is a well-established non-invasive measurement of subclinical atherosclerosis as well as an independent predictor for numerous CVDs [3,4,5].

Previous studies have explored the associations between IMT and single foods or nutrients [6,7] as well as with established dietary patterns [8]. A systematic review found that the Mediterranean diet may reduce IMT progression, especially in those participants with higher IMT, and that higher intakes of fruits and vegetables, wholegrains, soluble fibre and polyunsaturated fats as well as lower intakes of saturated fats were also associated with lower IMT [9].

We recently observed that a high intake of added sugar was associated with increased risk of stroke and coronary events in the Malmö Diet and Cancer Study (MDCS). However, while the consumption of sugar-rich foods was negatively associated with several CVDs, the intake of sugar-sweetened beverages (SSBs) was associated with increased risk of stroke [10]. Although a high sugar consumption has been suggested to be linked to the development of several CVDs and their risk factors, there is a lack of studies looking at the association between sugar consumption and prevalence of atherosclerosis. A cross-sectional study in asymptomatic adults found that high levels of carbonated SSB consumption were associated with a higher prevalence and degree of coronary artery calcium, a measurement for subclinical coronary heart disease [11]. However, a longitudinal study exploring the intake of different beverages found no association between SSB consumption and IMT [12]. Considering the limited support from previous studies, the aim of our study was to investigate the cross-sectional association between intake of different sugars and sugar-rich foods and beverages and IMT measured at the carotids in the MDCS.

## 2. Materials and Methods

### 2.1. Subjects and Data Collection

The data used in this study were obtained from a subcohort of the MDCS, a population-based cohort study. All men and women residing in Malmö (south of Sweden) who were born between 1923 and 1945 and 1923 and 1950, respectively, were invited to participate in the MDCS (*N* = 74,138) through a personal letter or an advertisement. The Ethical Committee at Lund University approved the MDCS (LU 51-90) and all the participants provided a written informed consent. The only exclusion criteria were limited knowledge of Swedish or mental incapacity. Data collection took place between 1991 and 1996, and it included assessment of dietary intake (see below), a self-administered lifestyle and socioeconomic questionnaire and anthropometric measurements taken by trained personnel at the research site. Ultimately, 30,446 participants (39% men) completed at least one part of the baseline assessment [13,14]. To perform further cardiovascular studies, 50% of the participants recruited between 1991 and 1994 were randomly selected to take part in the cardiovascular subcohort of MDCS (MDCS-CC) (*N* = 6103). The MDCS-CC participants underwent further tests, which included a study of the carotid arteries and the collection of fasting blood samples [15,16]. In this study, participants with ultrasound measurements of the carotids (*N* = 6057) and complete dietary, anthropometric and lifestyle information (*N* = 5711) were included. In addition, participants with previous history of CVD or diabetes mellitus were excluded from the present study sample. The main reason for excluding individuals with diabetes was the lower reported added sugar intake compared to individuals free of diabetes (7.5 vs. 10 %E, *p* = 9.7 × 10^−22^). Ultimately, the final sample amounted to 5269 participants (40% men) (Figure 1).

### 2.2. Dietary Data Collection

The MDCS’s modified diet history is a validated method for dietary data collection in this population [17,18]. The method consisted of three parts: (1) a 7-day food diary collecting information regarding prepared meals (lunch and dinner), cold drinks and supplement intake; (2) a food frequency questionnaire covering 168-items consumed regularly (breakfast, snacks and others not covered by the food diary) and hot drinks, the portion sizes were estimated with the help of a picture booklet with 4 portion sizes as reference for up to 48 food items; and (3) a 45–60 min interview with a trained interviewer, covering information about cooking methods and portion sizes of the items recorded in the food diary, and where the interviewer could check that there was no overlap in the information collected through the other two methods. The combined dietary data obtained was then introduced into a software where it was joint with the Malmö Food and Nutrient Database (based on Swedish Food Database PC KOST-93) [19,20]. An 18-day weighted food record was used for validation with a correlation coefficient for sugar of 0.60 for men and 0.74 for women [18].

### 2.3. Sugar Variables

Added sugars are sugars not naturally occurring in foods, i.e., which are added to foods during processing, manufacturing or preparation [21]. The variable added sugar intake was estimated for each participant by subtracting their intake of monosaccharides (mainly glucose and fructose) and sucrose from the main sources of naturally occurring sugars (fruits and berries, fruit juice and vegetables) from their total consumption of monosaccharides and sucrose [22]. The variable for free sugar intake (all added sugars plus those sugars naturally present in fruit juices [23,24]) was similarly estimated but without subtracting the sugar intake corresponding to fruit juices. Both variables were then categorised into six groups of non-alcoholic energy intake (%E) as follows: less than 5, 5–7.5, 7.5–10, 10–15, 15–20 and more than 20 %E from added or free sugar [22,25]. These categories were designed to be compared with existing recommendations for added or free sugar intake and to allow extreme intakes to be studied. The variable for total sugar intake (all monosaccharides and disaccharides present in the diet from any source [23]) was calculated by adding the total intake of monosaccharides (mainly glucose, fructose and galactose) and disaccharides (mainly sucrose, lactose and maltose). This variable was then categorised into four groups of non-alcoholic energy intake to reflect high and low intakes while still maintaining an adequate number of participants in each group: less than 15, 15–20, 20–25 and more than 25 %E from total sugar.

In addition, we examined the main sources of added sugar (treats, toppings and SSBs). Treats (i.e., pastries, sweets, chocolate and ice cream) covered food items that generally have a higher energy density as they have a high content of fat as well as sugar and that could more easily be consumed in large quantities (‘binge-eaten’). Toppings (i.e., table sugar, syrups, honey and jams) covered food items that owe their energy content mostly to a high amount of sugar alone and that are not so commonly binge-eaten. SSBs (i.e., soft drinks and fruit drinks, excluding pure fruit juice) covered added sugar sources presented in liquid form. The intakes (g/day) were then transformed into servings per week based on the definition of an average serving size according to the Swedish National Food Agency and food manufacturers. Thus, one serving of pastries (including cakes, pies, cookies and buns) = 60 g, sweets/candy = 60 g, chocolate = 60 g, ice-cream (including sherbet) = 75 g, table sugar (including syrup) = 10 g, jam (including marmalade, applesauce, jelly and honey) = 20 g and SSB = 280 g [22]. These variables were then categorized into groups as follows: treats as ≤2, >2–5, >5–8, >8–14 and >14 servings/week; toppings as ≤2, >2–7, >7–14, >14–28 and >28 servings/week and SSBs as ≤1, >1–3, >3–5, >5–8 and >8 servings/week [22].

### 2.4. Carotid Artery Measurements

To determine the IMT of the carotid arteries, qualified sonographers performed a B-mode ultrasound examination (Acuson 128CT system, Mountain View, CA, USA) at the right carotid artery. Three images were saved from the common carotid artery and three from the bifurcation of the carotids. IMT was measured over a 1 cm long segment, according to the leading-edge principle, off-line using the analysis program Artery Measurement System (AMS) [26,27]. All three images of the common carotid artery and the bifurcation were analysed. For IMT in the common carotid artery, the highest mean value was used (IMTcca), whereas the maximum value was used for IMT measured at the bifurcation (IMTbif) [28].

### 2.5. Other Variables

Information regarding the participants’ age and sex was obtained from the Swedish Registry via their personal identification number. The information regarding lifestyle factors, socioeconomic factors and medical history was recorded in a questionnaire that the participants filled out, including smoking habits (never, current or former smoker), educational level (elementary or less, primary and secondary school, upper secondary school, university education without a degree and university education with a degree), and leisure time-physical activity. Leisure-time physical activity, was categorised into five predefined groups based on Metabolic Equivalent Task (MET) hours per week: <7.5, 7.5–15, 15–25, 25–50 and >50 MET h/week) [29]. The information regarding alcohol consumption was compiled from the questionnaire and the food diary (zero consumers and quintiles of sex-specific consumption, described elsewhere [30]). Body Mass Index (BMI) (kg/m^2^) was calculated from height and weight measurements performed by a nurse during the participants visit at baseline [14,31]. Additionally, the collection of blood samples at baseline allowed for the ascertainment of a lipid profile for increased cardiovascular risk: high triglycerides (defined as ≥1.7 mmol/L or triglycerides lowering treatment), low high-density lipoprotein cholesterol (HDLc) (defined as < 1 mmol/L for men or <1.3 mmol/L for women) and high low-density lipoprotein cholesterol (LDLc) (defined as >4.1 mmol/L or lipid lowering treatment). Blood pressure was measured and hypertension was defined as a systolic blood pressure (SBP) of ≥130 mmHg or a diastolic blood pressure (DBP) of ≥85 mmHg or the use of antihypertensive drugs, as per the American Heart Association’s definition [32].

### 2.6. Statistical Analysis

Statistical analyses were performed using SPSS version 25 (IBM Statistics, New York, NY, USA) and the statistical significance level was established at *p* < 0.05, as per standard agreement. Added sugar intake was considered the main exposure variable, and free sugar, total sugar, treats, toppings and SSBs were secondary exposure variables. All the analyses were performed for men and women separately due to their different IMT levels and that cardiovascular risk factors and food habits differ between men and women. An ANOVA test was used to analyse continuous variables and a chi-square test was used for categorical variables for the two sexes separately and across sugar intake groups.

A general linear model was used to explore the association between the different sugar intake variables and the two measurements of IMT. Three different adjustment models were designed as follows: Model 1 was adjusted for age (years), start date, time between dietary data collection and IMT measurement (years) and season when the dietary data collection took place. Model 2 included model 1 and additional adjustments to cover lifestyle and dietary factors (alcohol consumption, leisure time physical activity, educational level, smoking habits, BMI, energy intake, coffee intake, meat intake, fruit and vegetable intake, fibre density and saturated fat intake). These variables were selected because they have been considered risk factors for atherosclerosis and are associated with sugar intake. Model 3 included model 2 and additional variables to cover known cardiovascular risk factors such as high triglycerides, high LDLc, low HDLc levels and the presence of hypertension [33]. These are factors mainly considered as mediators in the causal pathway between sugar intake and IMT. Logarithmically transformed IMT variables were used to achieve normal distribution. P_trends_ were determined using the categories of sugar intake (as %E or servings/week) as a continuous representation of the scale formed by the groups.

Sensitivity analyses were conducted excluding potential energy misreporters (*N* = 1019) and those participants who had reported dietary changes (*N* = 1233) in the year before baseline examinations from the main model for added sugar. Post hoc analyses were run to test the difference between the mean IMT measurements for those participants consuming more than 20 %E from added sugar and those with an intake of added sugar equal or below 20 %E for the main model of adjustment (Model 2).

## 3. Results

The study sample consisted of 5269 participants: 2114 men (40%) and 3155 women (60%). The mean age of the population was 57.3 years for both men and women. BMI was slightly higher for men (26.0 kg/m^2^) than for women (25.3 kg/m^2^). Mean values for IMT measurements were higher for men (IMTcca = 0.765 mm; IMTbif = 1.540 mm) than for women (IMTcca = 0.723 mm; IMTbif = 1.397 mm). The mean added sugar intake was 10.1 %E for men and 10.0 %E for women. The participants’ characteristics according to their added sugar intake are shown in Table 1. The highest representation for both men and women was for the subgroup that consumed between 10 and 15 %E from added sugar and the lowest was for the group reporting more than 20 %E from added sugar (Table 1). The majority of participants reported to consume less than one serving per week of SSBs.

Overall, no significant linear associations were found between any of the different types of sugar intake (added, free, or total), sugar-rich foods (treats or toppings), or beverages (SSBs) and any of the IMT measurements with any of our models of adjustment (Table 2 and Table 3, Appendix A). For example, in our main model (Model 2) for added sugar, P_trends_ for IMTcca were 0.81 for men and 0.98 for women and P_trends_ for IMTbif were 0.20 for men and 0.40 for women (Table 2). When analysing men and women together, our results remained non-significant (P_trend_ for IMTcca = 0.88 and P_trend_ for IMTbif = 0.12). In the sensitivity analyses, after excluding potential energy misreporters and those participants who had reported a substantial change in their diet before baseline, the results were unchanged for added sugar intake for both IMTcca and IMTbif (P_trends_ for IMTcca for men = 0.728 and for women = 0.518; and for IMTbif for men = 0.608 and for women = 0.420).

For both men and women, the highest IMTcca mean values seemed to be found amongst the group consuming more than 20 %E from added sugar (0.791 mm in men and 0.738 mm in women) (Table 2) compared to those with an added sugar intake below or equal to 20 %E (0.757 mm for men and 0.719 mm for women). In addition, there seemed to be a tendency of lower IMTbif values in those participants with added sugar intake above 20 %E (1.385 mm in men and 1.384 mm in women) (Table 2) compared to those with an added sugar intake below or equal to 20 %E (1.490 mm for men and 1.393 mm for women). In post hoc analyses, we compared the group consuming more than 20 %E from added sugar (2.1% of men and 1.7% of the women) with the groups consuming 20 %E or less; however, these differences were not significant (all *p* > 0.2).

## 4. Discussion

In our study, added sugar intake presented no significant linear association with IMT measured either at the common carotid artery or at the bifurcation of the carotids, nor did the intake of free or total sugar or sugar-rich foods and beverages.

Although sucrose, fructose and starch intakes have been found to have atherogenic effects in animal models [34,35,36], our study is one of the few to have investigated the association between sugar consumption and IMT in humans and the first to have explored added sugar intake in particular. Our findings are in line with a previous study that did not find any significant prospective association between SSB intake and IMTcca in women [12]. Other measurements of subclinical coronary artery disease have also been explored. For instance, a cross-sectional study on healthy Korean adults found that high intakes of carbonated SSBs were significantly associated with a higher prevalence and degree of coronary artery calcium [11]; a result that neither ourselves nor Wang et al. [12] were able to replicate when looking at the association between SSB consumption and IMT. Besides, a longitudinal study found that carbohydrate intake was positively associated with progression of atherosclerotic disease measured through IMT in postmenopausal women, particularly when the glycaemic index was high [37], while a Spanish study of asymptomatic adults found no significant association between glycaemic index or glycaemic load and IMT after a 1-year follow-up [38].

A high intake of added sugar is often part of an unhealthy food pattern characterized by lower intake of fibre, whole grains and fruit and vegetables [39]. Many individual studies have found associations between adherence to a healthy dietary pattern and lower IMT in different populations across the globe [40,41,42], although a systematic review and meta-analysis exploring several dietary patterns only found a non-significant trend between a higher adherence to healthier dietary patterns and lower IMT [8].

An increase in IMT is usually the first measurable symptom of atherosclerosis and carotid ultrasound measurements are considered the method of choice to gauge IMT progression and subclinical atherosclerosis [43]. Therefore, IMT can be considered a reliable marker for atherosclerotic risk, although it should not be considered as a risk factor per se, nor does it require any treatment [44]. The measurements included in our study (IMTcca and IMTbif) are both good predictors of cardiovascular outcomes [45]. In general, IMTcca values > 0.9 mm or above the 75th percentile are considered abnormal by the European Society of Cardiology and the American Society of Echocardiography, respectively [46,47]. However, IMT values are subject to age and sex differences, and therefore, there is no agreement within the scientific community as to where to establish the threshold for IMT abnormality [43,44,48,49]. MDCS participants are a relatively healthy population both compared to non-participants [14] and to other populations in terms of cardiovascular events [50]. The mean IMTcca values in our study were well below 0.9 mm (mean IMTcca for men = 0.759 mm, SD = 0.17 and for women = 0.719 mm, SD = 0.13) indicating that our population might be too healthy to ascertain the relationship between sugar intake and subclinical atherosclerosis.

The exploration of the major forms of sugar consumption (added, free and total) and the main sources of added sugar intake (treats, toppings and SSBs) is a great strength of our study. The selection of added sugar as our main exposure stems from the fact that these are sugars that are not naturally occurring in foods and beverages and, therefore, do not add any additional nutritional value other than an increase in energy intake. The consumption of SSBs has been the target of numerous studies mentioned in this paper, as this is a group that is easily defined and thus allows for comparison with our results. Both added sugar and SSBs have been the target of nutritional recommendations and policies targeting the reduction in sugar intake [23,51,52].

MDCS is a comprehensive dietary study with high-quality dietary data and validated data collection methods [17,18], which has allowed us to explore not only added sugar intake but also several sources of sugary foods and drinks and numerous risk factors. The slightly lower correlations for sugar intake found in validation studies [18] for men might translate into weaker associations in men compared to women in our sample.

It has been found that MDCS-CC participants suffering from diabetes have an increased risk of IMT progression at the bifurcation (IMTbif) but not on the common carotid artery (IMTcca) [28], however, it was also found that insulin resistance alone could not explain the development of atherosclerosis [15]. Additionally, participants with diabetes are usually recommended to limit sugar and caloric intake in their diets [53] as observed in our cohort and therefore might have introduced bias. Thus, it is a strength of our study that participants with diabetes were excluded from our study sample to reduce bias, even if this resulted in a reduction in statistical power. Other strengths of our study comprise the inclusion of several levels of adjustment, as well as sensitivity analyses excluding potential energy misreporters and those participants that have reported dietary changes prior to the baseline examination, which might reflect people with unstable dietary patterns and therefore less reliable data pertaining to their dietary intake. However, our cross-sectional study design does not allow to study the atherosclerotic progression nor establish causality. Therefore, the association between sugar consumption and IMT should also be explored in other populations, from different countries and different age groups, as well as using a longitudinal study design where the development and progression of atherosclerosis can be explored and a gradient of IMT can be established for each participant.

So, while the evidence between diet and subclinical atherosclerosis measured via IMT or other methods has been previously explored in the literature, the evidence behind a possible association between sugar intake, often associated with unhealthy dietary patterns, and IMT seems elusive. However, the tendency to higher IMTcca in the highest added sugar category observed in our sample coincides with previous findings in the MDCS, where the highest sugar intake was associated with a higher risk for certain CVDs [10]. These tendencies might point towards the need for further research, preferably in larger populations, with a higher added sugar intake and with a longitudinal design to confirm our results.

## 5. Conclusions

In conclusion, our study helps to fill in the gap of the understudied link between sugar intake and IMT, with a comprehensive analysis of the association between different forms of sugar intake as well as different consumption patterns of sugary foods and beverages and IMT as a marker for subclinical atherosclerosis in a large cohort study. However, we found no significant association between added sugar intake and IMT measured at the common carotid artery or the bifurcation of the carotids for men or women nor for other types of sugar intake (free or total sugar) or sugar-rich foods or beverages.

## Figures and Tables

**Figure 1 nutrients-13-01555-f001:**
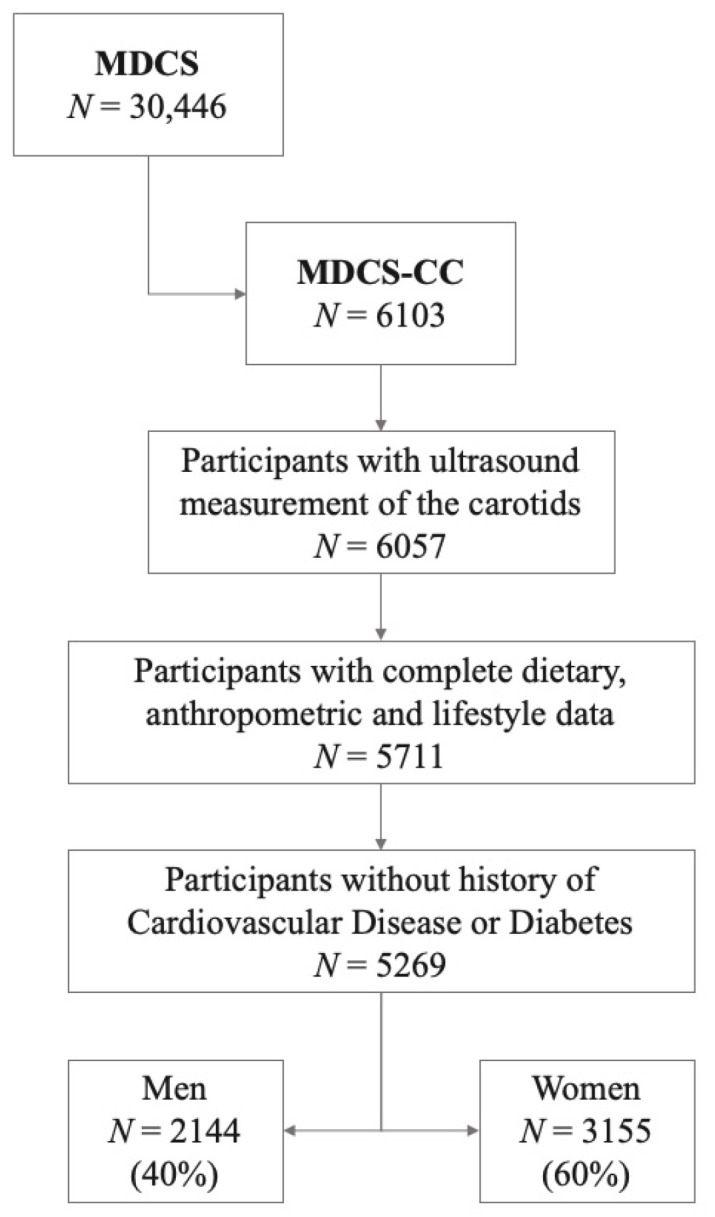
Flowchart of participant selection into the present study. (MDCS: Malmö Diet and Cancer Study; MDCS-CC: Malmö Diet and Cancer Study—Cardiovascular Cohort).

**Table 1 nutrients-13-01555-t001:** Baseline characteristics of the population in relation to added sugar intake (percentage on non-alcoholic energy intake, %E).

		Added Sugar Intake (%E)							
		<5	5–7.5	7.5–10	10–15	15–20	>20	Total	P_trends_
*N* (men/women)		198/277	389/621	557/810	731/1111	190/276	49/60	2114/3155	
Age (years) *	Men	56.0 (5.79)	56.6 (5.77)	57.5 (5.96)	57.5 (6.12)	58.1 (5.93)	57.8 (6.13)	57.3 (5.99)	<0.001
	Women	56.1 (5.79)	56.4 (5.79)	57.7 (5.98)	57.8 (5.98)	57.8 (5.60)	57.3 (5.87)	57.3 (5.92)	<0.001
BMI (kg/m^2^) *	Men	26.9 (4.07)	26.1 (3.11)	25.9 (3.24)	25.8 (3.40)	25.8 (3.12)	26.1 (4.10)	26.0 (3.38)	0.002
	Women	25.9 (4.57)	25.5 (4.37)	25.2 (3.80)	25.2 (4.04)	24.5 (3.87)	24.4 (4.01)	25.2 (4.09)	<0.001
Energy intake (kcal/day) *^1^	Men	2385 (722)	2528 (651)	2598 (632)	2700 (690)	2842 (739)	2766 (787)	2626 (688)	<0.001
	Women	1748 (512)	1903 (455)	1996 (478)	2084 (492)	2231 (554)	2304 (633)	2013 (508)	<0.001
Carbohydrates (%E) *	Men	40.6 (7.34)	41.6 (6.15)	44.2 (5.31)	46.3 (5.10)	49.7 (5.65)	53.0 (4.81)	44.8 (6.31)	<0.001
	Women	42.7 (7.01)	43.8 (5.85)	45.0 (5.40)	47.2 (5.42)	49.6 (5.21)	53.5 (5.30)	45.9 (6.05)	<0.001
Protein (%E) *	Men	17.6 (3.02)	16.5 (2.36)	15.7 (1.96)	14.7 (1.94)	13.5 (1.74)	12.3 (2.33)	15.4 (2.45)	<0.001
	Women	18.7 (3.02)	17.0 (2.40)	16.2 (2.25)	15.1 (2.09)	13.8 (1.84)	12.6 (2.25)	15.9 (2.63)	<0.001
Fat (%E) *	Men	41.7 (8.10)	41.8 (6.67)	40.2 (5.73)	39.0 (5.56)	36.7 (6.03)	34.7 (5.33)	39.8 (6.35)	<0.001
	Women	38.6 (7.50)	39.2 (6.58)	38.7 (5.85)	37.7 (5.78)	36.5 (5.58)	33.8 (4.99)	38.1 (6.17)	<0.001
Saturated fats (%E) *	Men	17.6 (5.26)	17.5 (4.25)	16.9 (3.66)	16.6 (3.67)	15.8 (3.96)	15.2 (3.18)	16.8 (4.00)	<0.001
	Women	16.3 (4.52)	16.7 (4.28)	16.6 (3.67)	16.2 (3.66)	16.1 (3.70)	14.9 (3.25)	16.4 (3.88)	0.001
Fibre (g/1000 kcal) *	Men	8.58 (3.07)	8.68 (2.72)	8.72 (2.47)	8.33 (2.25)	8.10 (2.59)	6.89 (1.98)	8.47 (2.53)	<0.001
	Women	11.3 (3.65)	10.5 (2.98)	10.0 (2.73)	9.64 (2.66)	8.74 (2.45)	7.95 (3.22)	9.94 (2.92)	<0.001
Fruits and vegetables (g/day) *	Men	390 (217)	356 (182)	382 (192)	351 (170)	343 (180)	272 (151)	361 (184)	<0.001
	Women	475 (205)	454 (203)	407 (167)	401 (174)	371 (169)	340 (224)	416 (184)	<0.001
Meat (g/day) *	Men	191 (91.1)	179 (75.6)	173 (75.2)	167 (68.7)	158 (65.3)	138 (71.5)	171 (74.4)	<0.001
	Women	118 (55.0)	118 (48.6)	116 (46.9)	112 (45.8)	107 (44.2)	104 (53.7)	114 (47.6)	<0.001
Fish (g/day) *	Men	46.0 (49.2)	48.2 (43.1)	46.9 (40.1)	41.0 (30.5)	38.6 (33.3)	30.9 (24.9)	43.9 (37.9)	<0.001
	Women	42.9 (31.5)	40.1 (29.2)	40.3 (29.8)	37.8 (27.2)	35.1 (28.7)	25.8 (23.5)	38.9 (28.9)	<0.001
Coffee (g/day) *	Men	612 (525)	569 (452)	533 (450)	539 (401)	553 (420)	626 (534)	553 (441)	0.233
	Women	556 (473)	529 (362)	494 (337)	504 (380)	474 (316)	636 (598)	511 (376)	0.094
SSBs (svg/wk) *	Men	0.43 (0.96)	0.65 (1.35)	1.30 (2.04)	2.74 (3.57)	6.01 (5.70)	10.8 (12.0)	2.24 (4.02)	<0.001
	Women	0.20 (0.47)	0.47 (0.93)	0.85 (1.35)	1.83 (2.55)	3.88 (4.18)	9.93 (9.31)	1.50 (2.90)	<0.001
Toppings (svg/wk) *	Men	2.20 (2.87)	4.91 (5.15)	8.95 (7.23)	15.6 (11.5)	26.8 (20.2)	32.7 (26.6)	12.0 (13.1)	<0.001
	Women	1.32 (1.56)	2.88 (2.68)	5.16 (4.05)	8.48 (7.18)	14.8 (12.8)	23.2 (21.1)	6.73 (8.04)	<0.001
Treats (svg/wk) *	Men	2.80 (2.46)	5.53 (3.81)	7.13 (4.42)	9.26 (5.89)	10.9 (8.00)	13.4 (12.9)	7.65 (5.96)	<0.001
	Women	3.00 (2.30)	4.88 (2.64)	6.63 (3.71)	8.65 (4.94)	11.2 (7.24)	13.1 (11.2)	7.20 (5.13)	<0.001
Time (years) *^2^	Men	0.76 (0.38)	0.80 (0.37)	0.79 (0.37)	0.77 (0.35)	0.75 (0.40)	0.74 (0.35)	0.78 (0.37)	0.231
	Women	0.84 (0.38)	0.81 (0.37)	0.82 (0.37)	0.81 (0.37)	0.78 (0.36)	0.78 (0.34)	0.81 (0.37)	0.093
HbA1c (%) *^3^	Men	4.82 (0.44)	4.82 (0.61)	4.76 (0.50)	4.78 (0.51)	4.82 (0.45)	4.86 (0.58)	4.79 (0.52)	0.818
	Women	4.82 (0.45)	4.80 (0.48)	4.82 (0.42)	4.83 (0.46)	4.78 (0.43)	4.78 (0.51)	4.81 (0.45)	0.910
High plasma glucose **^4^	Men	109 (63.4)	222 (61.0)	282 (55.7)	349 (51.2)	102 (59.0)	21 (46.7)	1085 (55.9)	0.007
	Women	96 (37.8)	202 (35.2)	262 (34.2)	381 (36.4)	81 (31.3)	21 (38.2)	1043 (35.3)	0.596
Alcohol, non-consumers **	Men	5 (2.5)	8 (2.1)	17 (3.1)	35 (4.8)	15 (7.9)	7 (14.3)	87 (4.1)	<0.001
	Women	16 (5.8)	35 (5.6)	46 (5.7)	84 (7.6)	24 (8.7)	5 (8.3)	210 (6.7)	<0.001
Alcohol, high consumers **^5^	Men	65 (32.8)	92 (23.7)	112 (20.1)	114 (15.6)	28 (14.7)	8 (16.3)	419 (19.8)	<0.001
	Women	50 (18.1)	110 (17.7)	127 (15.7)	145 (13.1)	27 (9.8)	6 (10.0)	465 (14.7)	<0.001
Sedentary **^6^	Men	15 (7.6)	37 (9.5)	42 (7.5)	65 (8.9)	15 (7.9)	8 (16.3)	182 (8.6)	0.060
	Women	35 (12.6)	51 (8.2)	70 (8.6)	99 (8.9)	22 (8.0)	6 (10.0)	283 (9.0)	0.617
University education **^7^	Men	24 (12.1)	45 (11.6)	83 (14.9)	80 (10.9)	16 (8.4)	6 (12.2)	254 (12.0)	0.185
	Women	46 (16.6)	103 (16.6)	104 (12.8)	110 (9.9)	16 (5.8)	3 (5.0)	382 (12.1)	<0.001
Current smokers **	Men	72 (36.4)	118 (30.3)	154 (27.6)	209 (28.6)	59 (31.1)	19 (38.8)	631 (29.8)	0.257
	Women	95 (34.3)	164 (26.4)	208 (25.7)	279 (25.1)	79 (28.6)	29 (48.3)	854 (27.1)	<0.001
High TG **^8^	Men	53 (30.6)	108 (29.7)	129 (25.5)	179 (26.3)	62 (35.6)	15 (34.1)	546 (28.1)	0.092
	Women	42 (16.6)	81 (14.1)	124 (16.1)	159 (15.1)	52 (20.2)	12 (21.8)	470 (15.9)	0.351
Low HDLc **^9^	Men	26 (15.3)	98 (27.2)	130 (25.9)	206 (30.4)	60 (34.9)	16 (35.6)	536 (27.8)	<0.001
	Women	57 (22.7)	175 (30.8)	215 (25.3)	605 (29.5)	99 (38.1)	21 (38.2)	872 (29.8)	0.004
High LDLc **^10^	Men	77 (45.8)	180 (51.3)	251 (50.9)	296 (44.3)	93 (55.4)	23 (52.3)	920 (48.6)	0.055
	Women	110 (44.0)	256 (45.3)	388 (51.5)	506 (49.4)	124 (48.6)	26 (47.3)	1410 (48.6)	0.189
Hypertension **^11^	Men	175 (88.4)	338(86.9)	4657 (83.8)	603 (82.5)	159 (83.7)	47 (95.9)	1789 (84.6)	0.043
	Women	200 (72.2)	452 (72.8)	620 (76.5)	859 (77.3)	209 (75.7)	45 (75.0)	2385 (75.6)	0.254
Season, winter **^12^	Men	55 (27.8)	124 (31.9)	175 (31.4)	233 (31.9)	54 (28.4)	13 (26.5)	654 (30.9)	0.123
	Women	88 (31.8)	203 (32.7)	234 (28.9)	352 (31.7)	75 (27.2)	15 (25.0)	967 (30.6)	0.351
Season, spring **^12^	Men	54 (27.3)	95 (24.4)	118 (21.2)	163 (22.3)	36 (18.9)	15 (30.6)	481 (22.8)	0.123
	Women	70 (25.3)	142 (22.9)	198 (24.4)	240 (21.6)	72 (26.1)	18 (30.0)	740 (23.5)	0.351
Season, summer **^12^	Men	29 (14.6)	29 (10.0)	64 (11.5)	88 (12.0)	21 (11.1)	11 (22.4)	252 (11.9)	0.123
	Women	43 (15.5)	91 (14.7)	103 (12.7)	137 (12.3)	37 (13.4)	8 (13.3)	419 (13.3)	0.351
Season, autumn **^12^	Men	60 (30.3)	131 (33.7)	200 (35.9)	247 (33.8)	79 (41.6)	10 (20.4)	727 (34.4)	0.123
	Women	76 (27.4)	185 (28.9)	275 (34.0)	382 (34.4)	92 (33.3)	19 (31.7)	1029 (32.6)	0.351

*N*: number of observations. Svg/wk: servings per week. * Presented: mean (standard deviation). ** Presented: number of observations (percentage of participants within group of added sugar intake). ^1^ Non-alcoholic energy intake. ^2^ Time: time in years between dietary data collection and IMT measurement (missing values for men = 164 and for women = 187). ^3^ HbA1c (missing values for men = 175 and for women = 199) ^4^ High plasma glucose: fasting plasma glucose > 5.6 mmol/L (missing values for men = 173 and for women = 202). ^5^ High alcohol consumers: more than 25.7 g/day for men and more than 14.0 g/day for women. ^6^ Leisure time and physical activity: Sedentary (<7.5 MET h/week). ^7^ Education: university degree or higher. ^8^ High triglycerides (TG): ≥1.7 mmol/L or treatment. ^9^ Low high-density lipoprotein cholesterol (HDLc): <1 mmol/L for men or <1.3 mmol/L for women. ^10^ High low-density lipoprotein cholesterol (LDLc): >4.1 mmol/L or treatment. ^11^ Hypertension: SBP ≥ 130 mmHg or DBP ≥ 85 mmHg or treatment. ^12^ Season: season when dietary data collection took place.

**Table 2 nutrients-13-01555-t002:** Association between added sugar intake (percentage on non-alcoholic energy intake, %E) and intima media thickness (mm) measured at the common carotid artery (IMTcca) and at the bifurcation of the carotids (IMTbif). IMT presented as mean value (95% confidence interval).

		Added Sugar Intake (%E)						
		<5	5–7.5	7.5–10	10–15	15–20	>20	P_trends_
IMTcca								
*N* * (men/women)		173/254	365/576	507/770	685/1052	175/261	45/55	
Model 1	Men	0.763 (0.738–0.787)	0.759 (0 742–0.777)	0.764 (0.749–0.779)	0.753 (0.740–0.765)	0.763 (0.738–0.787)	0.783 (0.736–0.831)	0.711
	Women	0.715 (0.700–0.731)	0.727 (0.716–0.737)	0.719 (0.710–0.728)	0.717 (0.710–0.728)	0.731 (0.716–0.746)	0.747 (0.715–0.780)	0.621
Model 2	Men	0.755 (0.729–0.782)	0.756 (0.737–0.776)	0.767 (0.750–0.783)	0.754 (0.740–7.769)	0.764 (0.738–0.790)	0.784 (0.735–0.832)	0.811
	Women	0.715 (0.698–0.732)	0.727 (0.715–0.739)	0.718 (0.708–0.729)	0.716 (0.707–0.726)	0.728 (0.711–0.745)	0.742 (0.707–0.776)	0.985
Model 3	Men	0.748 (0.719–0.777)	0.749 (0.728–0.771)	0.757 (0.738–0.776)	0.747 (0.730–0.764)	0.752 (0.724–0.779)	0.770 (0.720–0.820)	0.960
	Women	0.717 (0.699–0.734)	0.728 (0.715–0.741)	0.717 (0.705–0.729)	0.715 (0.704–0.726)	0.725 (0.707–0.743)	0.738 (0.704–0.773)	0.702
IMTbif								
*N* * (men/women)		108/159	270/376	345/514	498/712	120/191	35/42	
Model 1	Men	1.591 (1.475–1.708)	1.547 (1.472–1.621)	1.498 (1.431–1.565)	1.490 (1.435–1.545)	1.511 (1.400–1.622)	1.419 (1.216–1.622)	0.083
	Women	1.397 (1.312–1.482)	1.401 (1.346–1.457)	1.365 (1.317–1.413)	1.345 (1.305–1.386)	1.441 (1.364–1.519)	1.382 (1.217–1.548)	0.435
Model 2	Men	1.563 (1.435–1.691)	1.522 (1.438–1.605)	1.475 (1.399–1.550)	1.473 (1.408–1.538)	1.493 (1.373–1.612)	1.407 (1.198–1.616)	0.203
	Women	1.400 (1.308–1.492)	1.416 (1.353–1.479)	1.388 (1.333–1.443)	1.370 (1.320–1.420)	1.450 (1.364–1.537)	1.373 (1.200–1.546)	0.402
Model 3	Men	1.555 (1.416–1.693)	1.501(1.407–1.596)	1.459 (1.372–1.546)	1.455 (1.379–1.531)	1.482 (1.355–1.610)	1.375 (1.159–1.590)	0.207
	Women	1.399 (1.302–1.495)	1.423 (1.353–1.493)	1.395 (1.333–1.457)	1.376 (1.318–1.434)	1.442 (1.351–1.533)	1.369 (1.197–1.542)	0.300

*N*: number of observations. * Due to the lack of information regarding certain covariates in Model 3, the number of observations (*N*) is lower for Model 3, i.e., for IMTcca *N* = 168, 351, 493, 665, 168 and 44 for men and *N* = 250, 564, 752, 1023, 253 and 55 for women and for IMTbif *N* = 104, 259, 337, 485, 115 and 35 for men and *N* = 155, 371, 502, 700, 186 and 42 for women. Model 1: Adjusted for age, start date, time between baseline and IMT measurement and season. Model 2: Adjusted for model 1 plus alcohol consumption, leisure time—physical activity, education, smoking habits, BMI, energy intake, coffee, meat, fruits and vegetables, fibre and saturated fat. Model 3: Adjusted for model 2 plus high triglycerides, low HDLc, high LDLc and hypertension. Statistical significance established for *p* < 0.05.

**Table 3 nutrients-13-01555-t003:** Association between sugar-sweetened beverage intake (servings per week, svg/wk) and intima media thickness (mm) measured at the common carotid artery (IMTcca) and at the bifurcation of the carotids (IMTbif). IMT presented as mean value (95% confidence interval).

		Sugar-Sweetened Beverage Intake (svg/wk)					
		≤1	>1–3	>3–5	>5–8	>8	P_trends_
IMTcca							
*N* * (men/women)		1097/1862	369/632	195/239	133/137	156/98	
Model 1	Men	0.755 (0.744–0.765)	0.758 (0.741–0.775)	0.771 (0.748–0.794)	0.781 (0.753–0.809)	0.765 (0.739–0.790)	0.129
	Women	0.723 (0.717–0.729)	0.715 (0.705–0.724)	0.716 (0.701–0.732)	0.725(0.704–0.746)	0.742 (0.717–0.767)	0.643
Model 2	Men	0.755 (0.742–0.768)	0.761 (0.742–0.780)	0.768 (0.744–0.793)	0.779 (0.750–0.808)	0.760 (0.733–0.787)	0.311
	Women	0.723 (0.715–0.731)	0.712 (0.701–0.724)	0.714 (0.697–0.731)	0.720 (0.698–0.742)	0.736 (0.711–0.762)	0.914
Model 3	Men	0.745 (0.729–0.761)	0.750 (0.729–0.771)	0.764 (0.737–0.790)	0.774 (0.744–0.804)	0.748 (0.719–0.777)	0.257
	Women	0.722 (0.713–0.732)	0.714 (0.701–0.726)	0.710 (0.692–0.727)	0.717 (0.694–0.740)	0.733 (0.707–0.759)	0.622
IMTbif							
*N* * (men/women)		776/1234	264/432	135/169	91/88	110/71	
Model 1	Men	1.497 (1.452–1.543)	1.560 (1.485–1.635)	1.499 (1.395–1.604)	1.468 (1.341–1.595)	1.535 (1.420–1.650)	0.776
	Women	1.383 (1.352–1.414)	1.369 (1.317–1.420)	1.321 (1.239–1.404)	1.451 (1.337–1.565)	1.323 (1.196–1.450)	0.637
Model 2	Men	1.470 (1.414–1.527)	1.536 (1.452–1.619)	1.473 (1.363–1.583)	1.466 (1.334–1.599)	1.533 (1.411–1.655)	0.464
	Women	1.403 (1.361–1.445)	1.382 (1.323–1.441)	1.340 (1.253–1.428)	1.466 (1.347–1.585)	1.339 (1.207–1.472)	0.526
Model 3	Men	1.453 (1.382–1.524)	1.516 (1.423–1.610)	1.466 (1.346–1.585)	1.444 (1.304–1.584)	1.507 (1.375–1.639)	0.561
	Women	1.408 (1.357–1.460)	1.390 (1.325–1.454)	1.338 (1.246–1.431)	1.463 (1.342–1.584)	1.322 (1.188–1.456)	0.307

IMT presented as mean value (95% confidence interval). *N*: number of observations. * Due to the lack of information regarding certain covariates in Model 3, the number of observations (*N*) is lower for Model 3, i.e., for IMTcca *N* = 1065, 360, 186, 127 and 151 for men and *N* = 1816, 619, 234, 132 and 96 for women and for IMTbif *N* = 756, 258, 127, 88 and 106 for men and *N* = 1208, 426, 166, 86 and 70 for women. Model 1: Adjusted for age, start date, time between baseline and IMT measurement and season. Model 2: Adjusted for Model 1 plus alcohol consumption, leisure time—physical activity, education, smoking habits, BMI, energy intake, coffee, meat, fruits and vegetables, fibre and saturated fat. Model 3: Adjusted for Model 2 plus high triglycerides, low HDLc, high LDLc and hypertension. Statistical significance established for *p* < 0.05.

## Data Availability

The dataset presented in this article are not readily available because of ethical and legal restrictions. Requests to access the dataset should be directed to the Chair of the Steering Committee for the Malmö cohorts, see instructions at https://www.malmo-kohorter.lu.se/english (accessed on 7 July 2019).

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
