# Peer review of "Association between Sugar Intake and Intima Media Thickness as a Marker for Atherosclerosis: A Cross-Sectional Study in the Malmö Diet and Cancer Study (Sweden)"

_nutrients, 2021, doi:10.3390/nu13051555_

Round 1
Reviewer 1 Report
This study examines the associations between sugar intake and intima media thickness (IMT) in a large cohort of Swedish adults using data from the Malmö diet and cancer study. This paper adds to the literature on sugar and the development of atherosclerosis as measured by IMT. The authors recognize the limitations of this cross-sectional study while making effort to control for confounding variables in their analysis. Overall, this is an excellent paper that only needs minor revisions. I have a few suggestions that I believe would strengthen the paper.
Minor issues:
- Minor grammatical issues and misplaced commas throughout the paper, but these do not detract from the overall understanding
- Lines 48-50 – this sentence could be rewritten to make the “risk factors for them” (for CVDs?) clearer
- Lines 97-98 – the I would like to see some discussion (in the discussion section) of how these correlations for sugar could have potentially influenced the results, especially for men
- Line 167 – why was p<0.05 chosen? Simply because it is standard/typical?
- Section 2.5 Statistical analysis – was any analysis conducted without the potential confounding variables?
Strengths:
- Subjects and data collection – very well described
- Section 2.5 Statistical analysis – glad to see the various models described that were adjusted for potential confounding variables
- Lines 190-195 – Glad to see that sensitivity analyses were conducted
- Lines 307-308 – Glad to see recognition of the limitation of a cross-sectional design
Author Response
Thank you so much for all your comments and suggestions. Please see attached document for a detailed answer.

Reviewer 2 Report
Authors investigated association between sugar intake and IMT. Although the results are negative, but this topic is interesting for the prevention of CVD. I have some questions to develop this manuscript.
- Do authors have any information about HbA1c? Even after excluded patients with diabetic mellitus, HbA1c is an important factor to know participants’ background.
- According to study participants, at least half of them have dyslipidemia. How many people receive any medicine like statins? I think statins can influence IMT significantly and probably result in non-significance.
- Please show about factors of baseline characteristics which were used for adjusted Models. For example, start date, time between dietary data collection, season when the dietary data collection took place and lifestyle which were written from Lane 176 in Page 5.
- Please show the normal range of IMTcca and IMTbif in methods.
- Do authors have any discussion with many kinds of sugar intake? For example, total, added and free sugar intake or treats, toppings and SSBs according to Table 2-3 and Table S1-3. Which sugar intake is the most influential?
Author Response

(The authors gave the same response as above.)

Reviewer 3 Report
Excellent paper filling the gap between sugar consumption and development of atherosclerosis.
The objective of the study is investigating a crucial point in life-style cardiovascular prevention. The direct role of sugar intake in the development and progression of atherosclerosis is still debated. The study is based an a large sample of individuals from a well validated database, excluding individuals with known cardiovascular disease or diabetes. Carotid ultrasound measurements are appropriate to investigate subclinical atherosclerosis. All possible sources of sugar in the diet were investigated and categorized.
However, there is no mention of individuals with the condition termed "pre-diabetes" nor the association between IMT and glycemia.
About conclusions: although there was no direct association between sugar intake and carotid IMT, it must be considered that the study is cross-sectional with a young sample of individuals. More information could derive from a longitudinal study with older individuals.
Overall, the study is accurate and well written.
Author Response

(The authors gave the same response as above.)

Round 2
Reviewer 2 Report
Authors answered well. I agree to accept this manuscript.